# Identification and Characterization of a New Platinum-Induced TP53 Mutation in MDAH Ovarian Cancer Cells

**DOI:** 10.3390/cells9010036

**Published:** 2019-12-21

**Authors:** Ilaria Lorenzon, Ilenia Pellarin, Ilenia Pellizzari, Sara D’Andrea, Barbara Belletti, Maura Sonego, Gustavo Baldassarre, Monica Schiappacassi

**Affiliations:** Division of Molecular Oncology, Centro di Riferimento Oncologico di Aviano (CRO), IRCCS, National Cancer Institute, Via Franco Gallini 2, 33081 Aviano, Italy; ilaria.lorenzon@cro.it (I.L.); ilenia.pellarin@cro.it (I.P.); ilenia.pellizzari@gmail.com (I.P.); sdandrea@cro.it (S.D.); bbelletti@cro.it (B.B.); msonego@cro.it (M.S.)

**Keywords:** epithelial ovarian cancer, platinum resistant cells, TP53 secondary mutations, aberrant mitosis, multinucleated giant cells

## Abstract

Platinum-based chemotherapy is the therapy of choice for epithelial ovarian cancer (EOC). Acquired resistance to platinum (PT) is a frequent event that leads to disease progression and predicts poor prognosis. To understand possible mechanisms underlying acquired PT-resistance, we have recently generated and characterized three PT-resistant isogenic EOC cell lines. Here, we more deeply characterize several PT-resistant clones derived from MDAH-2774 cells. We show that, in these cells, the increased PT resistance was accompanied by the presence of a subpopulation of multinucleated giant cells. This phenotype was likely due to an altered progression through the M phase of the cell cycle and accompanied by the deregulated expression of genes involved in M phase progression known to be target of mutant TP53. Interestingly, we found that PT-resistant MDAH cells acquired in the TP53 gene a novel secondary mutation (i.e., S185G) that accompanied the R273H typical of MDAH cells. The double p53^S185G/R273H^ mutant increases the resistance to PT in a TP53 null EOC cellular model. Overall, we show how the selective pressure of PT is able to induce additional mutation in an already mutant TP53 gene in EOC and how this event could contribute to the acquisition of novel cellular phenotypes.

## 1. Introduction

Epithelial ovarian cancer (EOC) is the fifth leading cause of death for cancer in women [1]. Due to the lack of robust methods for early detection, most diagnoses are made when the cancer has already metastasized within the abdomen. Standard care for EOC patients combines radical surgery with carboplatinum-taxol chemotherapy. Despite a good rate of initial response, development of platinum (PT)-resistant recurrence is a frequent event and predicts poor prognosis for advanced EOC patients [2,3]. Thus, improving response to platinum represents an unmet goal in EOC treatment. After the initial molecular characterization performed with The Cancer Genome Atlas [4], different comprehensive genomic analyses of EOC have been done to unveil new oncogenic drivers and the molecular events linked to platinum resistance in the most common histotypes of EOC. These studies revealed a marked intertumoral and intratumoral heterogeneity with few recurrent somatic mutations in EOC [5,6,7] underscoring the need of new therapeutic targets that might also hit PT-resistant recurrences.

PT resistance has been linked to alterations in several processes such as drug transport, drug inactivation, DNA damage response, DNA repair, apoptosis, and autophagy [8,9,10,11]. Many studies have been done to identify genes and mechanisms directly associated to resistance to PT therapy. We have recently contributed to this issue reporting the selection and preliminary characterization of three new isogenic models of PT-resistant EOC cell lines—MDAH-2774, TOV-112D and OVSAHO. This work has pointed out a common ability of the three isogenic PT-resistant cells to resolve the PT-induced DNA damage compared to parental cells. Moreover, all PT-resistant cells displayed a change in their morphology and a higher ability to grow on mesothelium [12].

In the present work, we better characterized MDAH-2774 PT-resistant (PT-res) cells reporting the appearance of a novel mutation in TP53 induced by platinum pressure that is linked to the increased resistance to PT and an altered mitotic division observed in PT-res cells.

## 2. Materials and Methods

### 2.1. Cell Culture

MDAH-2774 (CRL-10303) parental and SKOV-3 (HTB-77) cells are from ATCC. Cisplatin-resistant (PT-res) isogenic cells were generated as described [12]. All cell lines were maintained in RPMI-1640 medium (Sigma-Aldrich Co., St. Louis, MO, USA) containing 10% heat-inactivated FBS, 100 U/mL penicillin and streptomycin (Sigma-Aldrich Co., St. Louis, MO, USA) at 37 °C in a 5% CO_2_ atmosphere. After the selection process, PT-res cells were kept in PT-free medium. MDAH-2774 PT-res clones were obtained from PT-res pools by plating them at 0.7 cell/well dilution in a 96-multiwell plate. All cell lines were authenticated in our lab using the Cell ID TM System (Promega, Madison, WI, USA) protocol and using Genemapper ID Ver 3.2.1 to identify DNA STR profiles.

### 2.2. Compounds and Drugs Treatment

Dose-response curves were performed essentially as previously described [13,14]. Briefly, EOC cells were seeded in 96-well culture plates and treated with increasing doses of cisplatin (CDDP) (TEVA Italia) for 72 h. For dose-response curves of p53-silenced MDAH, cells were seeded in 96-well culture plates and after 24 h transduced with sh-ctrl or sh-p53 lentiviral particles (Mission Sigma TRCN0000003756). Forty-eight hours after transduction, cells were treated with increasing doses of CDDP for 16 h and released with complete medium for 24 h. Cell viability was determined by MTS assay using the CellTiter 96 AQueous cell proliferation assay kit (Promega). 2′,7′-dichlorodihydrofluorescein diacetate (H2-DCF-DA) and cycloheximide (CHX) were acquired from Sigma. Double thymidine block was performed incubating cells with thymidine (2.5 mM) (Sigma).

### 2.3. Vectors and Transfections

The p53^R273H^ mutant was subcloned in the pEGFP-C1 vector (Clontech, Mountain View, CA, USA) and in the pLPC vector (provided by Dr R. Maestro) from the pCMV/Neo-p53^R273H^ (Addgene#16439) into the BamHI restriction site. The pEGFP-p53 and pLPC-p53 double mutant were generated using a QuikChange Site-Directed Mutagenesis Kit (Agilent, Santa Clara, CA, USA) using oligonucleotides carrying the indicated mutation. Plasmids were transfected in SKOV-3 cells using FuGENE HD Transfection Reagent (Promega) according to the manufacturer’s indications. Stable expressing clones were selected by culturing cells with G418 (0.5 mg/mL) (pEGFP) or puromycin (1 μg/mL) (pLPC). For lentiviral production, 293FT cells (Invitrogen, Carlsbad, CA, USA) were co-transfected, using a standard calcium phosphate precipitation with the lentiviral based shRNA constructs (pLKO) and lentiviral vectors pGag-Pol (Sigma) and pVSV-G (Invitrogen). Seventy-two hours after transfection, medium containing viral particles was used to transduce EOC cell lines. pLKO vectors encoding for control, and p53 (TRCN0000003756) shRNAs were purchased from Sigma.

### 2.4. Preparation of Cell Lysates, Immunoblotting, and Immunoprecipitation

Cell lysates were prepared using cold RIPA buffer (150 mM NaCl, 50 mM TRIS-HCl (pH 8), 0.1% SDS, 1% Igepal, and 0.5% NP-40) containing proteases inhibitor cocktail (Roche, St Louis, MO, USA), phosphatase inhibitors (1 mM Na_3_VO_4_ and 10 mM NaF, Sigma-Aldrich) plus 1 mM DTT. Protein concentrations were determined using the Bio-Rad protein assay (Bio-Rad, Hercules, CA, USA) Proteins were separated in 4–20% SDS–polyacrylamide gel electrophoresis (SDS-PAGE) (Criterion Precast Gel, Bio-Rad) and blotted onto a nitrocellulose membrane (Amersham, GE Healthcare). Immunoprecipitations were performed using 600 μg of cell lysate in HNTG buffer (20 mM HEPES, 150 mM NaCl, 10% glycerol, 0.1% Triton X-100) plus 1 μg of antiDNA-PKcs antibody (H-163 sc-9051, Santa Cruz Inc., Santa Cruz, CA, USA) and incubating overnight at 4 °C. The immunocomplexes were precipitated by adding protein A agarose-conjugated for additional 1 h and 30 min at 4 °C and finally separated on SDS-PAGE, for western blot analysis. Immunoblotting were performed using the following primary antibodies: DNA-PKcs (H-163 sc-9051, 1:1000), cyclin A (H-432 sc-751, 1:200), cyclin B1 (GNS1 sc-245, 1:200), phospho Rb (Ser780) (sc-12901, 1:500), stathmin (E-3 sc-55531, 1:1000), and p53 (DO-1 sc-126, 1:1000) were from Santa Cruz Inc.; phospho-p53 (Ser15) (#9284, 1:500), phospho-p53 (Ser37) (#9289, 1:500), phospho-H3 (Ser10) (D7N8E #53348, 1:500), Histone H3 (D1H2 #4499, 1:750), phospho-cdc2 (Tyr15) (10A11 #4539, 1:500) from Cell Signaling Technology; cyclin E2 (EP454Y ab40890, 1:1000), and NCAPH (ab154105, 1:300) from Abcam; DEPDC1 (1:500) (a kind gift from Prof G. Del Sal (Trieste University, Trieste, Italy); GAPDH (6C5 CB1001, 1:1000) was from EMD Millipore; α-tubulin (DM1A T9026, 1:5000) from Sigma-Aldrich; and CDK1 (C12720, 1:500) from BD Transduction Laboratories (Franklin Lakes, NJ, USA).

### 2.5. Protein Stability

p53 protein stability was evaluated in MDAH-2774 parental and PT-res clones (#12 and #42) treated with CHX (10 mg/mL) for the indicated time points using procedures as previously described in references [13,14].

### 2.6. RNA Isolation and Real-Time Polymerase Chain Reaction

Total RNA was isolated using TriZol reagent (Ambion, Waltham, MA, USA) following the manufacturer’s instructions. Two micrograms of total RNA were retro-transcribed using random hexamers and the AMV Reverse Transcriptase (Promega, Madison, WI, USA). Then, 1/20 of the obtained cDNAs were amplified using primers for the human FAM64A, CENPA, BUB1, NCAPH, C21orf45, DEPDC1, CCNE2, WDR67, CPSF6, and EPB41L [15,16]. Standard curves (10-fold dilution from 10−1 to 10−4 attomoles) were prepared and analyzed by quantitative reverse transcription polymerase chain reaction (qRT-PCR) using the CFX96 TM Real-Time PCR Detection System (Bio-Rad).

### 2.7. In Vitro Cell Proliferation Analyses

For growth curve analyses, MDAH-2774 parental and PT-res pools (pool#1 and pool#2) and SKOV-3 cells stably transfected with pLPC-p53^R273H^ or pEGFP-p53^R273H/S185G^ double mutant were cultured in six-well plates. Viable cells were counted daily in triplicate for five days by the trypan-blue dye exclusion method. PT-res clones (#12, #19, #42, and #51) were seeded in sextuplicate in 96-well plates and cell proliferation was determined daily (for five days) by MTS assay using the CellTiter 96 AQueous cell proliferation assay kit (Promega). All experiments were repeated at least three times.

### 2.8. Double Thymidine Block

To synchronize cells a double thymidine block was performed. Cells were plated in 60 mm dishes and after 24 h incubated with thymidine (2.5 mM) for 16 h. Cells were then washed with 1× phosphate-buffered saline (PBS) and incubated with fresh complete medium for 9 h before the second round of thymidine (2.5 mM) for 16 h. Synchronized cells were then washed with 1× phosphate-buffered saline (PBS), released with complete medium, and collected at the indicated time points for analysis of cell cycle by propidium iodide (PI)-DNA staining or analysis of protein by Western blot.

### 2.9. Immunofluorescence Analyses

For immunofluorescence staining, cells plated on coverslips were fixed in PBS 4% paraformaldehyde (PFA) at room temperature, blocked in PBS-1% bovine serum albumin (BSA), and permeabilized in PBS 0.2% Triton X-100. Stains were performed with primary antibodies α-tubulin-fluorescein isothiocyanate (FITC) (clone DH1A, F2168, 1:150) and γ-tubulin (clone GTU-88, T6557, 1:100) (Sigma Aldrich) and cleaved caspase 3 (Asp175) (#9661, 1:300) and phospho- histone H3 (Ser10) (D7N8E #53348, 1:300) from Cell Signaling Technology (Danvers, MA, USA) and p53 (DO-1 sc-126, 1:200) from Santa Cruz Inc. (Santa Cruz, CA, USA). Then samples were washed in PBS and incubated with secondary antibodies (Alexa-Fluor 488- or 568-conjugated anti-mouse or anti-rabbit antibodies; Invitrogen) for 1 h at room temperature. TO-PRO-3 iodide (Invitrogen, Carlsbad, CA, USA) were used to visualize nuclei and Alexa-Fluor 647-Phalloidin (Invitrogen) for F-actin staining. Coverslips were analyzed using the TCS-SP8 Confocal Systems (Leica Microsystems Heidelberg GmbH, Wetzlar, Germany) interfaced with the Leica Confocal Software (LCS) (version 3.5.5.19976, Wetzlar, Germany) or the Leica Application Suite (LAS) software (version 6.1.1, Wetzlar, Germany). At least 10 fields were scored for each cell population and experimental condition.

### 2.10. Reactive Oxygen Species (ROS) Analyses

To quantify the production of intracellular ROS, cells were plated in 60 mm dishes and treated or not with CDDP for 24 h. After PT-treatment, cells were incubated with H2-DCF-DA 2,5 μM in RPMI without phenol red for 30 min at 37 °C and 5% CO_2_. Cells were washed in PBS 1X and collected by trypsinization in 1 mL of PBS 1X and analyzed using the BD FACSCanto II flow cytometer. 2′,7′-dichlorodihydrofluorescein diacetate (H2-DCF-DA) is de-esterified intracellularly and turns to highly fluorescent 2′,7′-dichlorofluorescein upon oxidation. The H2-DCF-DA molecule was excited at 504 nm, and the emission at 529 nm was detected and recorded by the instrument. For each sample 20,000 events were analyzed, applying a gating strategy to exclude the debris. The percentages of positive cells for oxidized H2-DCF-DA were determined for each treatment condition, taking into account intrinsic autofluorescence levels. All the data were analyzed using BD FACSDiva software (version 8, Franklin Lakes, NJ, USA).

### 2.11. Cell Cycle Analyses

EOC cells were plated in 60 mm dishes and collected after 48 h for the exponentially growing condition analysis. For the synchronized cells condition, a double thymidine block was performed to arrest cells in the G1/S phase. Cells were then released and collected at the indicated time points for the cell cycle analysis. Once harvested, cells were washed with 1× PBS, fixed in ice-cold 70% ethanol (added drop-wise to the pellet while vortexing to minimize clumping), and stored at −20 °C overnight. Fixed cells were then washed in 1× PBS and re-suspended in propidium iodide (PI) staining solution (50 μg/mL PI + 200 μg/mL RNaseA, in 1× PBS). Stained cells were subjected to flow cytometry analysis with a BD FACScan™ or a BD FACSCalibur™ flow cytometer (Franklin Lakes, NJ, USA). Distribution of cells in G1, S and G2/M phases of the cell cycle was analyzed with ModFit LT™ software (version 5.0, Franklin Lakes, NJ, USA) and plots were realized with FlowJo software (version 10, Franklin Lakes, NJ, USA).

### 2.12. Next Generation Sequencing (NGS)

For TP53 sequencing of PT-res pool cells, genomic DNA was extracted from cultured cells using Maxwell 16 DNA purification kit (Promega). Then, 50 ng of genomic DNA was amplified with TruSeq Custom Amplicon kit (TSCA, Illumina, San Diego, CA, USA) specially designed for the targeted sequencing of TP53 (13 amplicons) among others. Libraries were run in an Illumina MiSeq instrument achieving a median coverage >1000 reads. Data were aligned to human reference genome hg18 and analyzed, after quality control, using Variant Studio and the IGV program, reporting only variants with a mutant allelic frequency (MAF) greater than 5%. We considered not only variants inside the coding sequence but also the ones inside the 5′-, 3′-untranslated regions (UTRs) and inside splicing regulatory elements [17] (http://p53.iarc.fr/TP53GeneVariations.aspx). Validation of NGS data was performed using the DNA Sanger sequencing method using the Applied Biosystems™ Sanger Sequencing Kit and the ABI3130xl instrument (Applied Biosystems, Waltham, MA, USA).

### 2.13. Statistical Analysis

All statistical analysis was done using Graphpad Prism 8 software, and appropriate methods were employed. Standard deviations were used for designating error bars on different cases as described in the text. Comparisons of two different groups were done with the Student’s *t*-test and multiple groups were compared using ANOVA test. *p*-value ≤ 0.05 was considered significant (* *p* ≤ 0.05, ** *p* ≤  0.01, *** *p*  ≤  0.001, **** *p*  ≤  0.001).

## 3. Results

### 3.1. PT-res Clones Show an Impaired Growth Rate 

To understand the mechanism underlying acquired PT-resistance, we have recently generated and partially characterized three PT-resistant isogenic EOC cell lines [12]. Further characterization of these PT-res cells pointed out that both MDAH-2774 (MDAH) and PT-res pools presented an impaired growth rate accompanied by a small increase of cell death in PT-res pools, likely not sufficient to completely explain the reduced growth rate of PT-res cells (Appendix A). To better understand why these cells grew slower than the parental counterpart, we proceed with a single cell cloning of MDAH PT-res pools and isolated 23 PT-res single-cell clones from the two resistant pools. All clones were confirmed to be PT-resistant (not shown). We selected four clones for a more complete characterization. Their cisplatin (CDDP) IC50 was 2.5- to 6.7-fold higher than the one of parental cells (Figure 1a) and also higher than the IC50 observed for the PT-res pools 1 and 2 (i.e., 11 and 12 µM, respectively) [12]. When proliferation was tested, all PT-res clones presented an impaired growth rate (Figure 1b).

### 3.2. PT-res Clones Present an Increased Number of Aberrant Mitotic Figures

The impaired growth rate of PT-res pools is somehow in contrast with our previous data indicating a similar cell cycle progression pattern between MDAH parental and PT-res pools in exponentially growing conditions (DNA content in FACS analysis) [12]. Interestingly, a closer examination of these cells by transmission microscopy and immunofluorescence unveiled the presence of large cells only in PT-res pools but not in parental. These enlarged cells were often multinucleated cells and positive for cleaved caspase 3 (marker of apoptosis) with an overall increase of cleaved caspase 3–positive cells in PT-res pools respect to parental cells (Appendix A).

MDAH PT-res cell clones also contained a subpopulation of larger, multinucleated cells not present in parental cells (Appendix A). Thus, we decided to investigate if an alteration in cell cycle progression could subtend to the appearance of this particular subset of cells in PT-res population. Cell cycle progression in exponentially growing PT-res clones showed small differences between parental and PT-res clones, with the latter having a slightly higher G2/M population and a slight decrease in S phase cells if compared to parental cells (Appendix A). Then, we used a double thymidine block to synchronize MDAH parental and PT-res clones that were then released for 12, 16, and 24 h, allowing the cells to complete one mitotic division as reported previously [15]. This analysis was quite striking and informative, showing that while the phosphorylation of pRB and the S phase marker Cyclin A and the G2/M marker Cyclin B1 were equally regulated in parental and PT-res clones (Figure 2a, first, second, and third panels), PT-res cell clones displayed an earlier and prolonged phosphorylation of Histone H3 (pH3 Ser10, a recognized marker of M phase), accompanied by a higher expression of the inhibitory phosphorylation Tyr 15-CDK1 (Figure 2a fourth and sixth panels, and pCDK1Tyr15/CDK1 quantification below last panel). 

FACS analyses of DNA content of synchronized cells confirmed, in the PT-res clones, the persistence of an increased G2/M population 24 h after release from double thymidine block, compatible with the observed increased expression of mitotic markers at this time point and also revealed the presence of a population of larger cells with high DNA content (Appendix A).

These data suggested that MDAH PT-res cells probably presented a mitotic defect that could explain the higher number of multinucleated cells and increased apoptosis. Based on these results, we next quantified the number of mitosis using the phospho Ser10 Histone H3 antibody (accepted marker of M phase cells) in immunofluorescence analysis in cells synchronized by serum starvation for 72 h and then released in complete medium for additional 24 h. This analysis revealed that the four PT-res clones presented an increased number of mitosis/field (Figure 2b and Appendix A) accompanied by an increased number of multinucleated cells (Figure 2c). The quantification of multinucleated cells/field evidenced significant differences for all clones with respect to parental cells and no significant differences among the different PT-resistant clones (Figure 2c and Appendix A). Considering that multinucleated cells could be the consequence of an altered mitotic division, we studied more in detail the morphology of mitotic cells in parental and PT-resistant clones using immunofluorescence coupled with confocal analysis and staining the cells for γ-tubulin, an accepted centrosome marker, α-tubulin to evidence the mitotic spindle, and TO-PRO-3 for DNA staining. These analyses demonstrated that PT-resistant clones presented an increased number of aberrant mitotic cells that represented more than 50% of all scored mitoses, mainly categorized as multi-centrosome cell divisions (Figure 2d and Appendix A). 

Interestingly, as observed in PT-res pools, PT-resistant clones were more positive than parental cells for the expression of cleaved caspase 3 (Appendix A) and the increase in cleaved caspase 3–positive cells paralleled the increase in the percentage of aberrant mitosis. 

Overall, the data collected so far suggested that defects in M phase progression accompanied the acquisition of the PT-resistant phenotype of MDAH and resulted in an increased number of multi-nucleated giant cells (MNGCs) and an increase in cleaved caspase 3–positive cells. Both these phenotypes could explain the lower growth rate of PT-res MDAH cells respect to the parental counterpart without a clear difference of cell distribution in the different phases of the cell cycle in FACS analyses, as observed previously [15].

It is interesting to note that a very recent report suggests that MNGCs could contribute to the chemoresistant phenotype of MDA-MB-231 breast cancer cells by increasing the production of Reactive Species of Oxygen (ROS) [18]. Accordingly, we observed that MDAH PT-res clones presented a higher percentage of ROS positive cells respect to parental cells both under basal condition and after CDDP treatment (Appendix A), supporting the possibility that, in MDAH cells, MNGCs contribute to the onset of PT-resistance.

### 3.3. p53^MUT^ Downstream Targets Are Differently Modulated in PT-res Clones

Based on the above results, we tried to understand why MDAH PT-res cells acquired a MNGCs population, and thus, we focused on the possible role of the tumor suppressor TP53, which plays a pivotal role in the control of M phase progression after therapy-induced DNA damage. Several reports suggest that cells lacking a functional TP53 enter mitosis even in the presence of a mutated DNA, especially when a mutated TP53 (p53^MUT^) is expressed [15,19,20]. Also, loss of p53 has previously been shown to promote abnormal cell ploidy, increase in pSer10 H3, and perturbed progression through M phase after the release from nocodazole-induced M phase arrest [21]. In fact, cells lacking wild type p53 functions escape cell cycle checkpoints and may execute mitosis even after DNA damage and chromosomal aberrations, leading to the generation of MNGCs [18,19,22].

This evidence supports the possibility that the presence of p53^R273H^ in MDAH cells could be relevant for the onset of the multinucleated phenotype in MDAH PT-resistant cells. Interestingly, we and others have demonstrated that p53^MUT^ specifically regulates the expression of genes implicated in the correct progression of M phase and cell division [15,16,23].

Using MDAH cells, we recently reported that the proper expression, stability and activity of p53^R273H^, regulated by stathmin/DNA-PK interaction, was necessary for the completion of a timely cell division and for the increased resistance to PT-induced cell death via the regulation of several mitotic regulators including BUB1, CENPA, C21orf45, and NCAPH [15], all belonging to the so-called p53^MUT^ signature [16]. Therefore, we first analyzed the expression of the 10 genes belonging to this p53^MUT^ signature in parental and PT-resistant clones. Among the tested genes, we observed a consistent increased mRNA expression of CCNE2 and DEPDC1 and a decreased mRNA expression for NCAPH and C21orf45 in PT-res clones when compared to parental cells (Figure 3a,b). No significant variations were detected in the mRNA expression of BUB1, CENPA, WDR67, CPSF6, FAM64A and EPB41 mRNA, although a tendency for the overexpression of CENPA and CPSF6 was noted (Appendix A).

All these genes play a key role in mitotic cell division. CCNE2 has been linked to endoreplication and genomic instability [24,25]. C21orf45, also known as MIS18A, is a kinetocore protein belonging to the Mis18 complex assembly that is crucial for CENPA deposition at the centromere [26,27]. DEPDC1 is highly expressed in M phase and its silencing caused mitotic defects [28,29,30,31] and NCAPH is a regulatory subunit of the condensin complex required for the conversion of interphase chromatin into condensed chromosomes [32,33,34]. Overall, their altered expression could participate in the definition of the aberrant mitotic regulation.

### 3.4. DNA-PK_CS_ Is Not Involved in the Regulation of p53^MUT^ Activity in PT-res Clones

These data supported the possibility that MDAH PT-res cells had a different p53^MUT^ activity and, based on our previous results, suggested a role for DNA-PK/stathmin in the regulation of p53^MUT^ in EOC cells [15], we first tested if the expression of p53, stathmin and/or DNA-PK proteins was altered in PT-res clones respect to parental cells. Yet, we found no significant changes in their expression except for a slightly reduction of stathmin levels when assayed in exponentially growing condition (Figure 3c).

Considering that DNA-PK plays a major role in p53^MUT^ regulation via phosphorylation of Ser 37 affecting its stability and/or its transcriptional activity, we evaluated p53 expression and phosphorylation in MDAH parental and PT-res cells treated with CDDP and released for 24 h. All tested PT-res clones presented a higher phosphorylation of Ser37 and Ser15 of p53 respect to parental cells both under PT-treatment and after the release, without significant changes in total protein levels (Figure 3d). In the same settings, no differences in the expression of stathmin and DNA-PK were observed (Figure 3d). However, this increased phosphorylation was not associated to an increased binding to DNA-PK neither to an increased p53 protein half-life, at least in basal conditions (Figure 3e,f).

### 3.5. PT-res Clones Gain a New TP53 Missense Mutation after Selection Process

We next analyzed the complete coding sequence of TP53 from genomic DNA extracted from MDAH parental cells and PT-res pools and clones by targeted NGS to confirm the reported presence of p53^R273H^. Surprisingly, we found that all PT-res clones (and pools, data not shown) presented an additional mutation c.553A>G in exon 5 that determines a substitution of Serine residue in pos.185 to Glycine (p.S185G) that was not present in parental cells and that accompanied the p.R273H, c. 818G>A in exon 8 (Figure 4).

The collected data supported the possibility that TP53 double mutation was involved, at least in part, in the acquisition of the resistant phenotype. To verify this hypothesis, we silenced TP53 in both parental and PT-resistant clones and then treated cells with CDDP for 16 h. Results showed that down modulation of p53 expression induced a marked cell death level per se, suggesting that MDAH cells are addicted to the expression of mutant TP53, rendering the evaluation of PT-sensitivity difficult to be evaluated in this experimental setting (Appendix A).

Based on these results, we used a different approach and evaluated if the p53^S185G/R273H^ mutant could play a role in some of the biological characteristics of the PT-res clones described above. In order to verify this hypothesis, we used the p53 null EOC cell line, SKOV3 in which we overexpressed the p53^R273H^ or the p53^S185G/R273H^ proteins. Although the two proteins were equally expressed, the double mutant p53^S185G/R273H^ expression had a higher phosphorylation levels of Ser 37 and, partially, of Ser 15, accompanied by a decreased expression of NCAPH (Figure 5a) mimicking what was previously observed in PT-res clones (Figure 3d). Importantly, the expression of p53^S185G/R273H^ increased the resistance to platinum of SKOV3 cells significantly better than the p53^R273H^ single mutant (Figure 5b) supporting a role of this double p53 mutant in the onset of PT-resistance of MDAH cells.

To understand if the increased resistance to platinum of SKOV-3 p53^S185G/R273H^ transfected cells was accompanied by an increased number of multinucleated cells, we evaluated the number of MNGCs in SKOV-3 cells transfected with p53^R273H^ or with p53^S185G/R273H^ by immunofluorescence analyses specifically looking at the number of multinucleated in cells expressing the p53 proteins. Results showed that p53^S185G/R273H^ markedly increased the number of multinucleated SKOV-3 cells (Figure 5c) respect to cells expressing p53^R273H^. The increased number of multinucleated cells was accompanied by changes in expression of NCAPH (down-modulation) and DEPDC1 (up-regulation) mRNAs similar to the one detected in PT-res clones (Figure 5d). On the contrary, the expression of CCNE2 was not modified by p53^S185G/R273H^ in this model (not shown), suggesting that context dependent regulation of gene expression exist. Nevertheless, p53^S185G/R273H^ expression reduced SKOV-3 cell growth when compared to cells expressing p53^R273H^ single mutant (Figure 5e), overall, recapitulating most of the phenotype observed in PT-res cells. 

## 4. Discussion

Here, we report that MDAH PT-res cells acquired, under the pressure of platinum selection, a secondary mutation (the S185G missense substitution) in TP53 that accompanied the already present R273H pathogenic missense substitution. This p53^S185G/R273H^ double mutant was present in all tested clones and pools. Yet, while the R273H substitution is homozygous in MDAH cells the S185G seems to be heterozygous. p53^S185G^ is still categorized as a mutation of unknown significance in the literature and has been reported in the Cosmic database only in one case of ovarian endometrioid carcinoma and one case of esophageal carcinoma. Interestingly, the analyses of p53 binding domain 3D structure revealed that S185 and R273 are juxtaposed, supporting the possibility that this alteration has a specific significance in the context of mutant TP53. Intriguingly, in both cases p53^S185G^ substitution was found in tumors with known pathogenic TP53 mutations that alter the aminoacids H168 and K164 in the case of esophageal (COSP14113) [35] and W53 and W146 in the case of ovarian endometrioid carcinoma (COSP13878) [36]. It is therefore possible that the S185G mutation confers a specific advantage to a p53^MUT^ protein.

To understand if p53^S185G^ appeared as the selection of an already existing missense mutation or as a platinum-induced mutation, we deeply sequenced the parental MDAH cells (median coverage depth for TP53 was 5600X) in search of possible existing subclones carrying this substitution. Our results suggest that MDAH parental cells did not present any other p53 mutation except for the p53^R273H^ already described, implying that the pressure of PT selection induced the appearance of this co-occurring mutation.

The p53^S185G^ mutation was associated with higher pS37 phosphorylation not only in PT-res clones but also when transfected in SKOV3 cells, suggesting that it might confer increased activity to an already mutated TP53. How S185G substitution affect the pS37 phosphorylation is something that should be more deeply investigated in future work, although we could likely exclude that it could be due to an increased association between TP53 and DNA-PK_CS_.

The ectopic expression of p53^S185G/R273H^ protein had functional consequences since it significantly increased the resistance to PT in SKOV-3 cells and was associated with PT-resistance in MDAH cells. Based on the phenotypes observed in MDAH PT-res cells, we hypothesized that the more active p53^S185G/R273H^ protein could be partially responsible for the bypass of the G2/M checkpoints normally activated upon DNA damage by regulating mitotic regulators. This altered regulation eventually resulted in a deregulated cell division with the appearance of large multinucleated giant cells (MNGCs). Although we partially proved this point using the SKOV-3 model, we cannot verify the effective contribution of TP53 mutation on the phenotypes observed in MDAH cells, since these cells seem to be addicted to mutant TP53 expression. Therefore, more experiments and more appropriate study models are needed to definitively prove this point.

MNGCs could be generated after radiation or chemotherapy exposure [18,22,37,38]. MNGCs likely represent senescent, slowly growing cells that remain viable and secrete soluble signaling factors including ROS collectively known as senescence-associated secretory phenotype (SASP), in which TP53 plays crucial roles. In particular it has been proposed that SASP suppresses or promotes tumorigenesis depending on the TP53 status [39]. It is worth noting that an apoptosis-independent function of caspase 3 have been recently reported, indicating that caspase 3 could also facilitate DNA damage–induced genomic instability and carcinogenesis mediating the secretion of pro-survival factors [40,41,42,43]. This notion suggest that in our model cleaved caspase 3 expression could also be linked to a pro-survival function in PT-res cells since it has been already described as a molecule characterizing the SASP phenotype of MNGCs [22,40,43].

The role of TP53^MUT^ in the generation of MNGCs is still unclear, but since it has been proposed that MNGCs could represent an adaptive response of apoptosis-reluctant cells, TP53^MUT^ could play a pivotal role in allowing aberrant mitotic cells to complete the M phase and eventually survive as MNGCs. Interestingly, others and we demonstrated that TP53^MUT^ specifically regulates the transcription of genes involved in mitotic division [15,16], supporting the possibility that it confers additional surviving possibilities to cells with aberrant mitosis otherwise committed to die for mitotic catastrophe. The deregulation of some of these genes (i.e., CCNE2, DEPDC1, NCAPH and C21orf45) observed in PT-res MDAH clones is in line with this possibility.

In any case, converging evidences indicated that MNGCs ultimately contribute to the generation of progeny with stem cell–like properties and may have a fundamental role in cancer recurrences and chemoresistance [22,38]. This phenotype was also observed in several studies performed on ovarian cancer models [44,45,46,47,48]. Future studies are, however, necessary to properly clarify the molecular mechanisms underlying the formation of MNGCs and how TP53^MUT^ contribute to this phenotype. 

It is worth noting that we have observed and described changes in cell morphology and adhesion capabilities as characteristic shared by all the different EOC PT-res cells (i.e., MDAH, OVSAHO and TOV-112D) isolated in our lab, suggesting that the pressure of platinum selection might favor the appearance of PT-res clones with an unbalanced regulation of their epithelial and mesenchymal status [12]. Yet, the selection process we used to isolate PT-res cells induced the appearance of a subpopulation that could be classified as MNGCs producing high ROS levels in MDAH cells. In OVSAHO and TOV-112D PT-resistant cells, we did not observe defects in cell proliferation and caspase-3 activation (data not shown), suggesting that this could be a context dependent effect. We sequenced TP53 in both OVSAHO and TOV-112D cells and, while we confirmed the presence of the mutation reported in the literature (R342* and R175H, respectively), we did not observe in these cells the appearance of the additional p53^S185G^ (not shown), indirectly supporting its involvement in the typical phenotype of MDAH PT-res cells. The other molecular alterations that could contribute with TP53^MUT^ to the generation of MNGCs in MDAH are still unclear. However, the gene set enrichment analysis (GSEA) of MDAH, OVSAHO, and TOV-112D PT-res cells we recently performed evidenced that only MDAH PT-res cells had alteration in pathways controlling cell cycle progression and cell division [12].

## 5. Conclusions

Overall, our data demonstrated that platinum chemotherapy could induce additional mutations in TP53 and that the expression of double mutant p53^S185G/R273H^ could participate in the generation of PT resistance by regulating the progression through mitosis. Future more detailed works will be necessary to specifically unveil the underlying molecular mechanisms.

## Figures and Tables

**Figure 1 cells-09-00036-f001:**
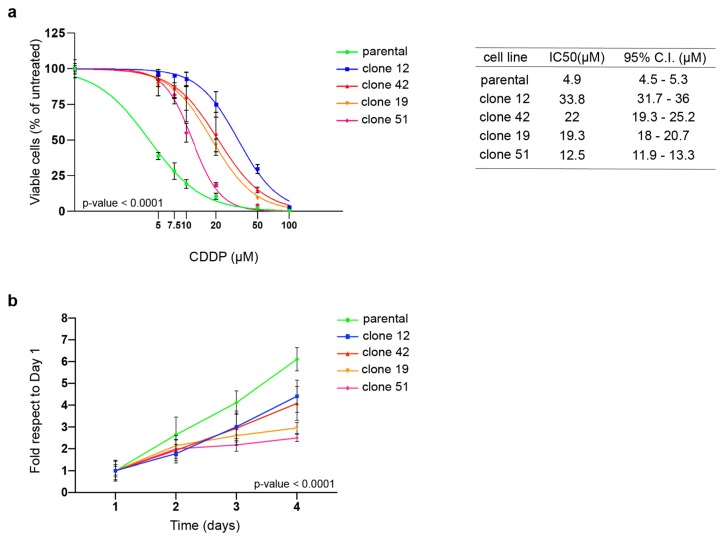
MDAH-2774 (MDAH) platinum-resistant (PT-res) clones show increased resistance to platinum (PT) and an impaired growth rate. (**a**) Nonlinear regression analyses evaluating cell viability of MDAH parental and PT-res clones treated with increasing doses of cisplatin (CDDP) for 72 h. Data are expressed as percentage of viable cells respect to the untreated cells and are the mean (±SD) of three biological replicates. The table on the right reports the IC50 and the confidence interval (CI) of each cell clone. Fisher’s exact test was used to calculate the global *p*-value reported in the graph. (**b**) Growth curves analyses of cells described in (**a**). Data are expressed as fold increase respect to day 1. Global statistical significance was determined by two-way ANOVA test.

**Figure 2 cells-09-00036-f002:**
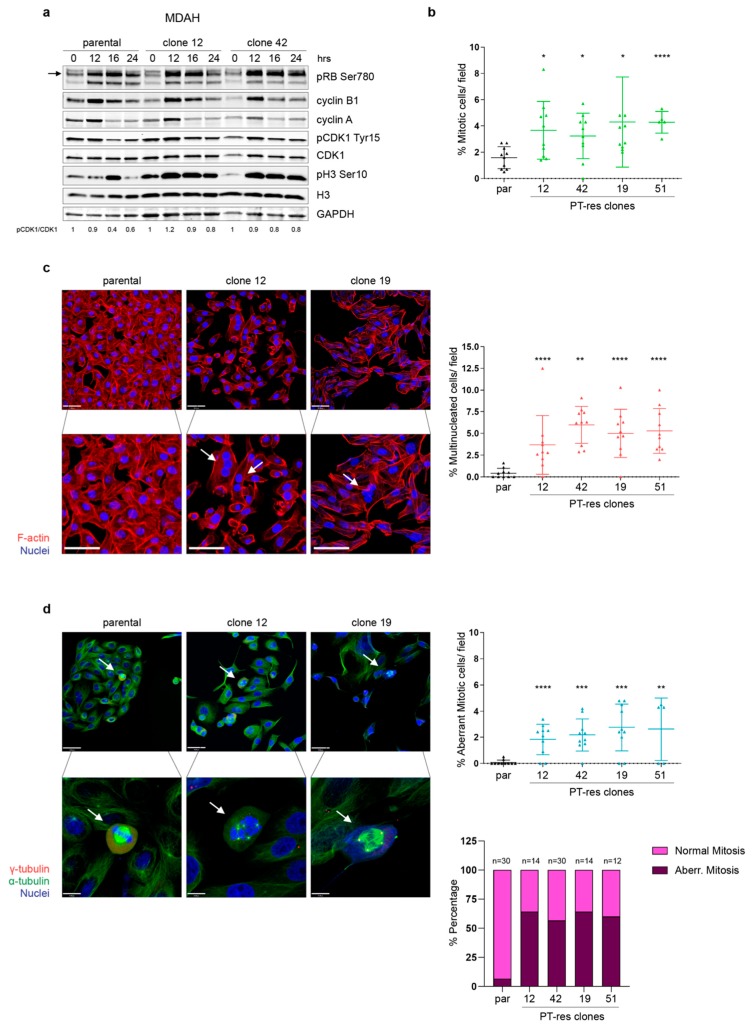
PT-res clones had an increased number of aberrant mitotic figures. (**a**) Western Blot analysis evaluating the expression of the indicated cell cycle markers in MDAH parental and PT-res clones (#12 and #42) synchronized by double thymidine block and released for the indicated hours (hrs). GAPDH was used as loading control. The quantification of pCDK1Tyr 15/CDK1 is reported below. (**b**) The graph reporting the number of mitotic cells/field (mean ± SD) in the indicated cells evaluated by immunofluorescence analyses using pS10 Histone H3 as a mitotic marker. (**c**) Representative images of MDAH parental and PT-res clones stained for F-actin (Phalloidin, red) and Nuclei (TO-PRO-3, pseudo colored in blue). In the higher magnification panels, white arrows point to multinucleated cells. Scale bar = 44 μm. On the right, the number of multinucleated cells/field is reported (mean ± SD). (**d**) Representative images of MDAH parental and PT-res clones, stained for α-tubulin (green), γ-tubulin (red) and TO-PRO-3 (nuclei, blue). Scale bar = 44 μm. In the lower panels, white arrows point to mitotic cells. Scale bar = 11 μm. On the right, the upper graph reports the number of aberrant mitosis/field (mean ± SD), and the lower graph presents the fraction of aberrant and normal mitosis expressed as percentage of total mitosis. Numbers above each column indicate the total number of mitosis analyzed for each cell type. Statistical significance was determined by a two-tailed unpaired Student’s *t*-test (* *p* < 0.05 and ** *p* < 0.01 and *** *p* < 0.001 and **** *p* < 0.0001).

**Figure 3 cells-09-00036-f003:**
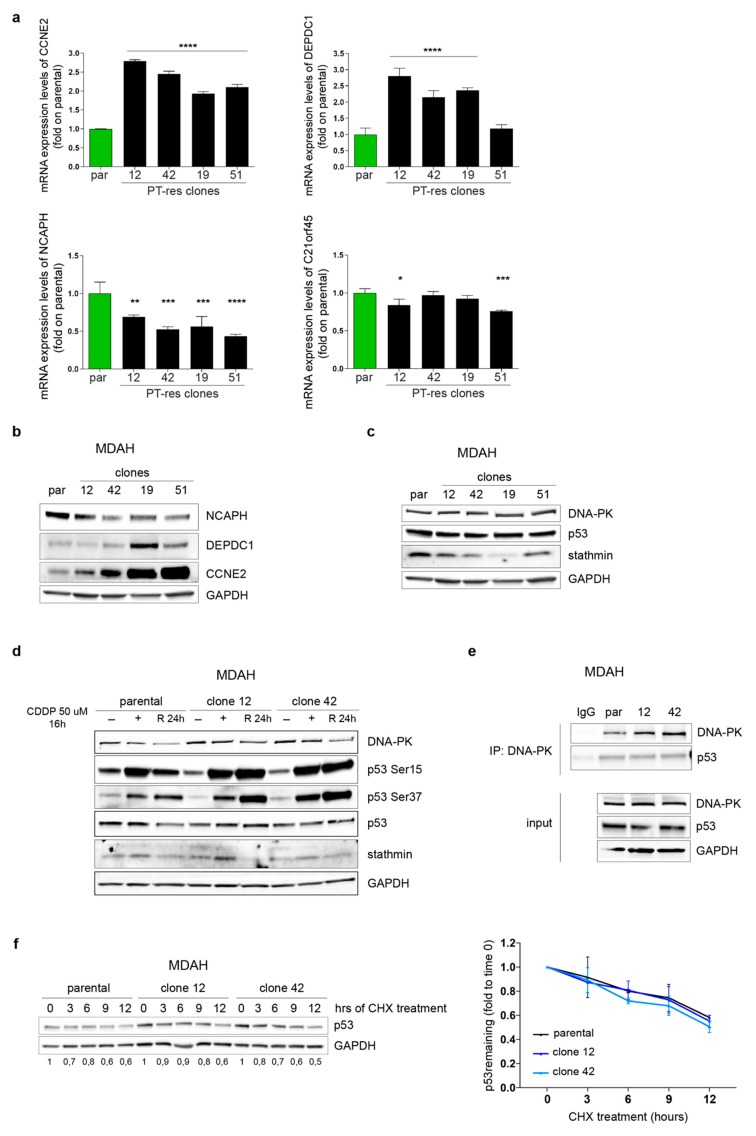
p53^MUT^ downstream targets are differently expressed in PT-res clones (**a**) Graphs reporting the normalized (to GAPDH) expression of the indicated genes in MDAH parental and PT-res clones evaluated by qRT-PCR analyses. Data are expressed as fold of mRNA expression in PT-res cells respect to parental cells and represent the mean (±SD) of at least of three independent experiments. Statistical significance was determined by one-way ANOVA test; a multiple comparison analysis was done to determine significant differences among groups (* *p* < 0.05 and ** *p* < 0.01 and *** *p* < 0.001 and **** *p* < 0.0001). (**b**,**c**) Western Blot analysis evaluating the expression of NCAPH, DEPDC1, and CCNE2 (**b**) and DNA-PK, p53, and stathmin (**c**) in parental and PT-res clones, as indicated. (**d**) Western Blot analysis evaluating the expression of DNA-PK, total and phosphorylated (Ser15 and Ser37), p53 and stathmin in parental and PT-res clones (#12 and #42) not treated (−) or treated with CDDP for 16 h (+) and allowed to repair for 24 h (R24h). (**e**) Co-immunoprecipitation analyses of endogenous DNA-PK and p53 proteins in parental and PT-res single clones (#12 and #42). Input shows the expression of the indicated proteins in the correspondent lysates; IgG represents the control IP using an unrelated antibody. (**f**) Western blot analysis of p53 expression in parental and PT-res single clones (#12 and #42) treated with cycloheximide (CHX) for the indicated times. The graph on the right reports the p53 expression as remaining fraction respect to T0 (mean ± SD of three different experiments), normalized respect to GAPDH expression. In the figure GAPDH was used as loading control.

**Figure 4 cells-09-00036-f004:**
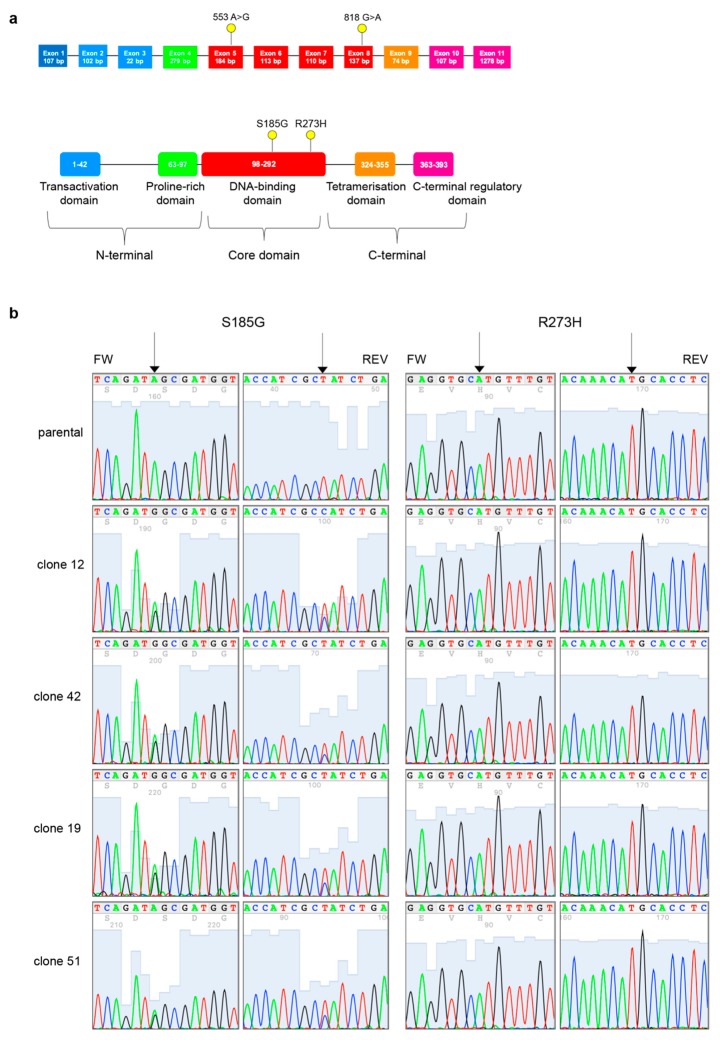
PT-res MDAH clones gain a new TP53 missense mutation. (**a**) Schematic diagram of p53 gene (top) and protein (bottom). Exons are color-coded according to the corresponding protein functional domains (transactivation domain in blue, proline-rich domain in green, DNA-binding domain in red, tetramerization domain in orange, and C-terminal regulatory domain in pink). Yellow dots depict the mutations detected in MDAH PT-res cells. (**b**) Representative four-color fluorescence electropherograms of p53 Sanger sequencing performed on parental and PT-res clones, black arrows indicate residues 273 and 185.

**Figure 5 cells-09-00036-f005:**
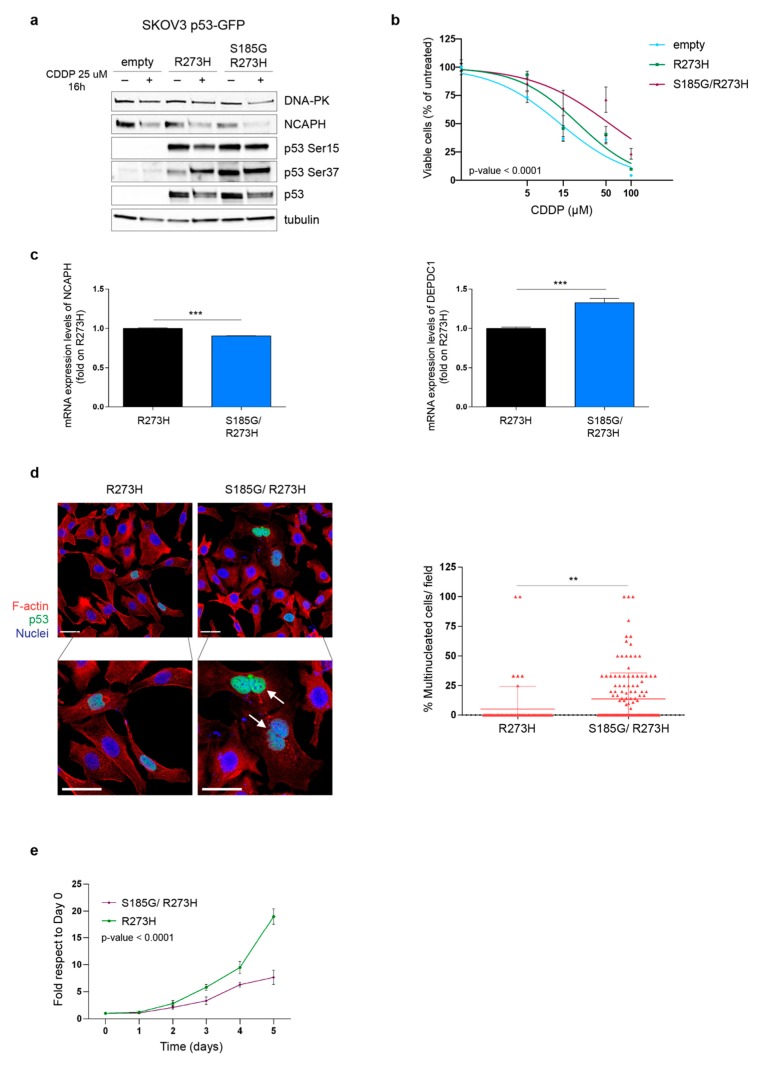
p53 ^S185G R273H^ expression confers resistance to PT-induced death (**a**) Western blot analyses evaluating the expression of DNA-PK, NCAPH, phosphorylated p53 (Ser15 and Ser37) in SKOV-3 cells stably transfected with pEGFP-p53^R273H^ or pEGFP-p53^S185G/R273H^ double mutant. Cells were treated or not with CDDP for 16 h. Tubulin was used as a loading control. (**b**) Nonlinear regression analyses of cell viability assay of cells described in (**a**) and treated with increasing doses of cisplatin (CDDP) for 72 h. Data are expressed as percentage of viable cells respect to the untreated cells and represent the mean (±SD) of 3 biological replicates. Fisher’s exact test was used to calculate the global *p*-value reported in the graph. (**c**) Representative images of SKOV-3 cells transfected with p53^R273H^ or p53^S185G/R273H^ double mutant stained for F-actin (red) and p53 (green). Nuclei are pseudo colored in blue (TO-PRO-3). Bottom panels show the indicated zoomed area for each condition (2× zoom) in which white arrows indicate multinucleated cells. Scale bar = 44 μm. On the right, the number of multinucleated cells counted among p53 positive cells/field is reported (mean ± SD). The entire coverslips were counted. (**d**) qRT-PCR analyses evaluating the expression of NCAPH and DEPDC1 in SKOV-3 cells transfected with p53^R273H^ or p53^S185G/R273H^. Data are expressed as fold of mRNA expression in double mutant expressing cells respect to single mutant expressing cells and represent the mean (±SD) of three independent experiments. In (**c**,**d**), statistical significance was determined by a two-tailed unpaired Student’s *t*-test (** *p* < 0.01 and *** *p* < 0.001). (**e**) Growth curves of SKOV-3 cells stably transfected with p53^R273H^ or p53 ^S185G/R273H^ double mutant. Data are expressed as fold increase with respect to day 0. Global statistical significance was determined by two-way ANOVA test.

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
