# Peer review of "Identification and Characterization of a New Platinum-Induced TP53 Mutation in MDAH Ovarian Cancer Cells"

_cells, 2019, doi:10.3390/cells9010036_

Round 1

Reviewer 1 Report

Polyploid giant cancer cells (multinucleated giant cells) have been observed in many cancers for over a century.  It has also been reported that chemotherapy and radiotherapy can result in the enrichment of multinucleated giant cells in several cancers, including ovarian cancer. But mechanisms of formation and role of these cells in the initiation and progression of human tumors is largely undefined.

In this study, Ilaria Lorenzon et al. demonstrated that a new platinum-induced TP53 mutation may play a role in the formation of multinucleated giant cell and acquired resistance to platinum.

Although the most phenotypes of the multinucleated giant cell have been reported (Int J Cancer. 2014 Feb 1; 134(3): 508–518, Oncogene. 2014 Jan 2; 33(1): 10.1038/onc.2013.96.), the function of double mutants of p53(S185G/R273H) is interesting. However, more experiments are needed to confirm their conclusion.

The authors overexpressed double mutant p53(S185G/R273H) in SKOV3 cells to investigate its role in the biological characteristics of the PT-resistant clones. However, the author's data suggested that this novel mutation may be specifically involved in the phenotype of MDAH PT-res cells, not in other cells. To further confirm the role of the double mutant p53 (S185G/R273H) in MDAH PT-res cells: 

1) The authors should Knockout (or Knockdown) p53 in their selected clones to confirm whether p53 (the mutated p53) is required for the PT-resistance and other phenotypes.

2) The author should perform the Knockout-Rescue experiments (Knockout endogenous TP53, then re-expression TP53 single or double mutant) in MDAH PT-res cells to confirm the functions of single and double mutations of p53.

Author Response

Reviewer 1

Polyploid giant cancer cells (multinucleated giant cells) have been observed in many cancers for over a century.  It has also been reported that chemotherapy and radiotherapy can result in the enrichment of multinucleated giant cells in several cancers, including ovarian cancer. But mechanisms of formation and role of these cells in the initiation and progression of human tumors is largely undefined.

In this study, Ilaria Lorenzon et al. demonstrated that a new platinum-induced TP53 mutation may play a role in the formation of multinucleated giant cell and acquired resistance to platinum.

Although the most phenotypes of the multinucleated giant cell have been reported (Int J Cancer. 2014 Feb 1; 134(3): 508–518, Oncogene. 2014 Jan 2; 33(1): 10.1038/onc.2013.96.), the function of double mutants of p53(S185G/R273H) is interesting. 

We thank Reviewer 1 for His/Her general appreciation of our study.

However, more experiments are needed to confirm their conclusion. The authors overexpressed double mutant p53(S185G/R273H) in SKOV3 cells to investigate its role in the biological characteristics of the PT-resistant clones. However, the author's data suggested that this novel mutation may be specifically involved in the phenotype of MDAH PT-res cells, not in other cells. To further confirm the role of the double mutant p53 (S185G/R273H) in MDAH PT-res cells: 

1) The authors should Knockout (or Knockdown) p53 in their selected clones to confirm whether p53 (the mutated p53) is required for the PT-resistance and other phenotypes.

2) The author should perform the Knockout-Rescue experiments (Knockout endogenous TP53, then re-expression TP53 single or double mutant) in MDAH PT-res cells to confirm the functions of single and double mutations of p53.

We perfectly agree with Reviewer 1. Knockdown/knockout of TP53 combined with the rescue of p53 single and double mutant is the most appropriate way to verify whether p53 is required for the establishment of the PT-resistant and MNGCs phenotypes. Unfortunately, as we now shown in the new supplementary Figure S6, the silencing of TP53 results in MDAH massive cell death, suggesting that these cells are addicted to the expression of mutant TP53. This evidence prevents the possibility to properly verify the role of p53 (S185G/R273H) on the response to platinum of these cells. Nevertheless, we tried to treat parental and PT-res silenced cells with CDDP using shorter period of drug exposure (i.e 16 hours) but the results are clearly masked by the mortality induced by the gene silencing. As requested, we added these data in this revised version of our manuscript (see new Figure S6) and we discuss this evidences in the discussion section (lines 461-463).

Reviewer 2 Report

Lorenzon and colleagues report the identification of a platinum-induced TP53 mutation regarding S185 in MDAH ovarian cancer cells, which was acquired under the pressure of platinum selection in a previous study.  Although the mutation was already reported in one case of ovarian endometrioid carcinoma and one case of esophageal carcinoma, it is still categorized as a mutation of unknown significance. The author show that the p53S185G/R273H double mutation resulted in a subpopulation of multinucleated giant cells and an altered progression through the M phase of the cell cycle. Overexpression of p53S185G/R273H in EOC cells lacking PT53 further resulted in increased resistance to platinum and phosphorylation of Ser37.

Despite these highly interesting findings, the following issues should be considered prior to publication:

Lines 190-193: The authors state that the observed impaired growth rate of PT-res pools is in contrast with their previous data indicating a similar cell cycle progression pattern parental and resistant MDAH cells. The authors may provide additional DNA content measurements of the specific clones analyzed in the present study to further validate potential defects in M phase progression of distinc PT-res clones, as suggested in line 244. In addition, are there differential effects between cells comprising p53S185G/R273H double and p53R273H single mutations regarding cell cycle progression?

Lines 207-208: The authors state “a higher expression of the inhibitory phosphorylation Tyr 15-CDK1 (Figure 2a fourth and sixth lanes)” in PT-res cell clones. However, the protein amounts in the respective western blot seem to be quite similar to those of the parental cells. The authors should add a quantitative analysis here to demonstrate the proposed effect.

Lines 274-276: The authors show increased mRNA expression of CCNE2 and DEPDC1 and a decreased mRNA expression for NCAPH and C21orf45 in PT-res clones in comparison to parental cells. Did the authors investigate mitotic regulators specifically altered in terms of the novel p53S185G/R273H double mutation, which differ in expression and/or activity from those known to be present in p53R273H single mutation cells?

Figure 5, lines 340-342: Although not mentioned in the results section, the amount of phosphorylated levels of Ser37 seems already highly elevated in cells overexpressing p53S185G/R273H but not treated with CDDP in comparison to untreated cells overexpressing only p53R273H. Could the authors comment on this observation?

Lines 312-320, 340-341: The authors observed an increased phosphorylation of Ser37 and Ser15 of p53 in PT-res cells compared to parental cells. However, this increased phosphorylation was not associated to elevated binding to DNA-PK or increased p53 protein half-life. Likewise, overexpression of p53S185G/R273H led to a higher phosphorylation level of Ser 37 and increased resistance to platinum. The authors should suggest a mechanism how the p53S185G mutation affects phosphorylation of Ser 37 and platinum resistance and investigate this mechanism experimentally in more detail.

Author Response

Lorenzon and colleagues report the identification of a platinum-induced TP53 mutation regarding S185 in MDAH ovarian cancer cells, which was acquired under the pressure of platinum selection in a previous study.  Although the mutation was already reported in one case of ovarian endometrioid carcinoma and one case of esophageal carcinoma, it is still categorized as a mutation of unknown significance. The author show that the p53S185G/R273H double mutation resulted in a subpopulation of multinucleated giant cells and an altered progression through the M phase of the cell cycle. Overexpression of p53S185G/R273H in EOC cells lacking PT53 further resulted in increased resistance to platinum and phosphorylation of Ser37.

Despite these highly interesting findings, the following issues should be considered prior to publication:

We thank Reviewer 2 for finding of high interest of our study.

Lines 190-193: The authors state that the observed impaired growth rate of PT-res pools is in contrast with their previous data indicating a similar cell cycle progression pattern parental and resistant MDAH cells. The authors may provide additional DNA content measurements of the specific clones analyzed in the present study to further validate potential defects in M phase progression of distinc PT-res clones, as suggested in line 244. In addition, are there differential effects between cells comprising p53S185G/R273H  double and p53R273H single mutations regarding cell cycle progression?

We thank Reviewer 2 for this comment that allow us to better clarify this point. The new data included in figure S2b-c, confirm that PT-res pools had a higher percentage of G2/M cells, especially when synchronized by double thymidine block. These new data are in complete agreement with our previous publication showing minor differences between MDAH parental and PT-Res evaluated in exponentially growing condition (2017 Sonego et al).

Lines 207-208: The authors state “a higher expression of the inhibitory phosphorylation Tyr 15-CDK1 (Figure 2a fourth and sixth lanes)” in PT-res cell clones. However, the protein amounts in the respective western blot seem to be quite similar to those of the parental cells. The authors should add a quantitative analysis here to demonstrate the proposed effect.

As requested we have now added the quantitative analysis of phosphorylation Tyr 15 CDK1 in the revised version of Fig 2a. 

Lines 274-276: The authors show increased mRNA expression of CCNE2 and DEPDC1 and a decreased mRNA expression for NCAPH and C21orf45 in PT-res clones in comparison to parental cells. Did the authors investigate mitotic regulators specifically altered in terms of the novel p53S185G/R273H double mutation, which differ in expression and/or activity from those known to be present in p53R273H single mutation cells?

We thank Reviewer 2 for this comment. We have investigated the expression some of these mitotic regulators in SKOV3 cells over expressing p53R273H and the novel double mutation p53S185G/R273H. In particular, the new Figure 5d shows that DEPDC1 and NCAPH mRNA expression is differentially modulated by p53R273H and p53S185G/R273H in SKOV-3 cells. However, we also evaluated in the same cells CCNE2 expression showing no difference, suggesting that context dependent effects in gene regulation by mutant TP53 exist. These data are now included in the manuscript and commented at lines 422-428. We have also completed the WB analysis in Figure 5a adding the expression of NCAPH and p53 pSer15.

We also show that p53S185G/R273H increased the number of multinucleated cells and decreased the growth of SKOV-3 cells (new figure 5c and 5e), overall supporting the possibility that p53S185G/R273H is at least partially responsible of the phenotypes observed in PT-res MDAH cells. This possibility is now commented in the discussion section (lines 459-461).  

Figure 5, lines 340-342: Although not mentioned in the results section, the amount of phosphorylated levels of Ser37 seems already highly elevated in cells overexpressing p53S185G/R273H but not treated with CDDP in comparison to untreated cells overexpressing only p53R273H. Could the authors comment on this observation?

We agree with Reviewer 2 on this point. We believe that the forced overexpression of TP53 combined with higher production of ROS observed in cells expressing p53S185G/R273H could result in higher DNA damage and explain the higher expression of phosphorylated p53 in untreated cells. Moreover, we looked at the expression also of Ser 15 in the same cells and in this case the major differences between p53R273H and p53S185G/R273H were observed in CDDP-treated cells as expected (See new figure 5a). Thus, although we cannot exclude that it is largely dependent of the overexpression approach used, this point certainly needs to be better clarified in future studies.

Lines 312-320, 340-341: The authors observed an increased phosphorylation of Ser37 and Ser15 of p53 in PT-res cells compared to parental cells. However, this increased phosphorylation was not associated to elevated binding to DNA-PK or increased p53 protein half-life. Likewise, overexpression of p53S185G/R273H  led to a higher phosphorylation level of Ser 37 and increased resistance to platinum. The authors should suggest a mechanism how the p53S185G mutation affects phosphorylation of Ser 37 and platinum resistance and investigate this mechanism experimentally in more detail.

The analyses of TP53 structure (see below an example of the 3D structure of p53 DNA binding domain) showed that S185 is juxtaposed to R273. Therefore, we believed that the substitution of S185G, at least in the context of p53R273H mutant protein, could alter its 3D conformation and capacity to bind the DNA eventually resulting in higher phosphorylation and/or activity. We added this consideration in the discussion section as requested (see lines 437-439).

In the figure is reported the 3D structure of TP53 DNA binding domain in which on the left is marked the position of R273 (green dot) and on the right the position of S185 (blue dot).

Please see attached File

Reviewer 3 Report

This manuscript by Lorenzon et al. describes the characterisation of a number of novel cisplatin-resistant ovarian cancer cell clones of the MDAH -2774 parental cell line, all of which are shown to have acquired a secondary p53 mutation (S185G). The authors further show that overexpression of R273H/S185G p53 mutant in SKOV3 p53-null cells increases resistance to cisplatin. PT-resistant MDAH-2774 clones are also shown to have mitotic aberrations/dysregulated S10P H3 and evidence is presented of increased multinucleated cells/endoreplication which the authors hypothesise may be linked to the secondary p53 mutation.

Overall, this is a well presented manuscript with some interesting findings and high quality data that would be of interest to the readership of Cells. One limitation is that is presently unclear whether the effects on mitosis/multinucleation etc reported in the PT-resistant MDAH-2774 clones are due, or partly due, to the secondary mutation of p53 (for example whether these effects can be partially recapitulated by the p53 double mutant in the SKOV3 - any data should be shown). Another outstanding but more difficult question is whether these effects are important in conferring cisplatin resistance. Both these points restrict the impact of the manuscript a little.

There are several points that should be addressed as detailed below in a revision before the manuscript would be suitable for publication in Cells.

Points to address:-

1)  Fig. 1b and Fig.S1a - the impaired growth rate presented here could be due to reduced cell proliferation or increased cell death or a combination of both of these - this could be more explicitly explained in the accompanying text (lines 191-197) especially in light of the increased positive staining for apoptotic marker cleaved caspase 3 in the PT-res pools. Growth curves were performed by trypan blue dye exclusion - what was the % of dead cells at each time point in the parental versus PT-res pools, are basal levels higher in the PT-res pools? - this could be presented alongside.

2) Multinucleation is very nicely presented in Fig 2c through phalloidin/DAPI staining but is much less obvious on phase contrast images. For Fig. S1b can you show similar zoomed in images - of higher resolution so that the multinucleated cells can be clearly seen and check careful positioning of arrows. Comment on the altered morphology of the majority of the PT-res cells compared to parental cells?  

3) Fig. S1c, show higher resolution, zoomed in images so co-localisation of CC3 staining with multinucleated cells can clearly be seen.

4) In Fig. S2, for clone 42, 34% multinucleated giant cells are reported - show cell cycle profile compared to parental - increased abnormal ploidy should be visible ie cells of greater than normal G2/M DNA content?

5) Show cell cycle profiles (DNA histograms rather than % bar chart of cells in different phases) after double thymidine block to show effective synchronisation by block and compare profiles for parental cells and clones at 12, 16 and 24h after release (in main or

supplementary data).

6) lines 205 & 208 - you are referring to ‘panels’ rather than ‘lanes’ - replace

7) Fig. 2a - pRB Ser780 - multiple bands are visible - which one is S780? Molecular weight markers?

8) Fig 2a  - pH3 should be H3?

9) lines 244-249 - are you meaning ref 12? It is a little surprising that you may not be seeing discernible effects on the cell cycle - is this because of the low number of MNGCs in the pools/some clones? If progression in M phase is impaired - i.e. if it takes longer - would a higher number of M phase cells not be expected?

10) Section 3.3 - lines 254-267, using isogenic p53 wild type and p53-null cancer cell clones, loss of p53 has previously been shown to promote abnomal cell ploidy, elevated S10P H3 and perturbed progression through M phase after release from nocodazole-induced arrest (Allison & Milner, 2003, Cancer Research 63:6674-9) - this study is relevant and should be cited.

11) comment on role of S15P p53 given sustained levels observed in Fig. 3d with cisplatin washout in PT-res clones cf. parental cells.

12) Following overexpression of the R273H/S185G p53 double mutant in the SKOV3 cells compared to control or R273H cells, is a reduction in growth rate observed? Similarly, have the authors analysed whether there any effects on mitotic fidelity? Is there any change in expression of CCNE2? i.e. the data using the SKOV3 indicates that the secondary mutation of p53 promotes PT-resistance, however, mechanistically is this double mutation causally associated in any way to the mitotic-related observations made in the first half of the manuscript - is there any supportive data from the SKOV3 transfections that can be presented that shows effects similar to those in the MDAH P-res clones? This would add value to the manuscript and the strength of conclusions that can be made eg .lines 422-423

13) Comment on pathophysiological relevance - does this secondary mutation in p53 occur in vivo in ovarian cancer or any other cancers?

Author Response

Reviewer 3

This manuscript by Lorenzon et al. describes the characterisation of a number of novel cisplatin-resistant ovarian cancer cell clones of the MDAH -2774 parental cell line, all of which are shown to have acquired a secondary p53 mutation (S185G). The authors further show that overexpression of R273H/S185G p53 mutant in SKOV3 p53-null cells increases resistance to cisplatin. PT-resistant MDAH-2774 clones are also shown to have mitotic aberrations/dysregulated S10P H3 and evidence is presented of increased multinucleated cells/endoreplication which the authors hypothesise may be linked to the secondary p53 mutation.

Overall, this is a well-presented manuscript with some interesting findings and high quality data that would be of interest to the readership of Cells. One limitation is that is presently unclear whether the effects on mitosis/multinucleation etc reported in the PT-resistant MDAH-2774 clones are due, or partly due, to the secondary mutation of p53 (for example whether these effects can be partially recapitulated by the p53 double mutant in the SKOV3 - any data should be shown). Another outstanding but more difficult question is whether these effects are important in conferring cisplatin resistance. Both these points restrict the impact of the manuscript a little.

We thank Reviewer 2 for His/Her general appreciation of our study and for highlighting the high quality of presented data.

There are several points that should be addressed as detailed below in a revision before the manuscript would be suitable for publication in Cells.

Points to address:

1)  Fig. 1b and Fig.S1a - the impaired growth rate presented here could be due to reduced cell proliferation or increased cell death or a combination of both of these - this could be more explicitly explained in the accompanying text (lines 191-197) especially in light of the increased positive staining for apoptotic marker cleaved caspase 3 in the PT-res pools. Growth curves were performed by trypan blue dye exclusion - what was the % of dead cells at each time point in the parental versus PT-res pools, are basal levels higher in the PT-res pools? - this could be presented alongside.

As Reviewer 3 suggested we have calculated and reported in the new figure 1b the percentage of dead cells at each time point evaluated by trypan blue exclusion cell. These data have now been explained in lines 206 and 267.

2) Multinucleation is very nicely presented in Fig 2c through phalloidin/DAPI staining but is much less obvious on phase contrast images. For Fig. S1b can you show similar zoomed in images - of higher resolution so that the multinucleated cells can be clearly seen and check careful positioning of arrows.  Comment on the altered morphology of the majority of the PT-res cells compared to parental cells?  

As suggested, we have included in the new Figure S1c a magnification of the phase contrast field.

3) Fig. S1c, show higher resolution, zoomed in images so co-localisation of CC3 staining with multinucleated cells can clearly be seen.

Thank you for this suggestion. We have now included the magnification of the caspase staining in the new figure S1c.

4) In Fig. S2, for clone 42, 34% multinucleated giant cells are reported - show cell cycle profile compared to parental - increased abnormal ploidy should be visible ie cells of greater than normal G2/M DNA content?

As shown below in the FACS analyses of DNA content, included in this point by point response for Reviewer evaluation only, in clone 42 it is possible to identify a population of larger cells with higher DNA content (green population). This analysis however suffers of the limit of FACS analyses in which is hard to distinguish between real giant multinucleated cells and cells doublets due to inaccurate FACS separation. For this reason, although these data fully confirm the presence of giant multinucleated cells in clone 42 we included in Figure S2 only the cell cycle distribution of the blue population (see figure below, P3 gate) that could be undoubtably assigned to G1, S and G2/M population.

In the figure is reported the FACS analyses of DNA content of parental and clone 42 cells. The left panels show cell size as determined by forward and side scatter fluorescence. The right panels show the FL2 Area and width fluorescence give by PI that highlight the distribution of DNA content. In red is depicted the population of larger cells with a high content of DNA. In blue the population analyzed by FACS and reported in figure S2. In red cells dead cells or cells that cannot be properly classified with this approach and therefore excluded from the analyses.

5) Show cell cycle profiles (DNA histograms rather than % bar chart of cells in different phases) after double thymidine block to show effective synchronisation by block and compare profiles for parental cells and clones at 12, 16 and 24h after release (in main ors upplementary data).

As requested, in the new Figure S2c we have now included the FACS analyses of DNA content in parental and PT-res MDAH cells that underwent to double thymidine block and then released in thymidine free medium for up to 24 hours. These data are discussed in the result section at lanes 262-265 of the revised manuscript.

6) lines 205 & 208 - you are referring to ‘panels’ rather than ‘lanes’ – replace

We regret this error. We have replaced the correct word in this revised manuscript.

7) Fig. 2a - pRB Ser780 - multiple bands are visible - which one is S780? Molecular weight markers?

We have modified the Figure 2a adding an arrow that identifies the band corresponding to pS780.

8) Fig 2a  - pH3 should be H3?

We thank Reviewer 3 and apologized for this mistake. The revised version of our manuscript now presents a correct Figure 2a.

9) lines 244-249 - are you meaning ref 12? It is a little surprising that you may not be seeing discernible effects on the cell cycle - is this because of the low number of MNGCs in the pools/some clones? If progression in M phase is impaired - i.e. if it takes longer - would a higher number of M phase cells not be expected?

We agree with Reviewer 3 and thank Him/Her for this comment that allow us to better explain this point.

Our previous results (indeed ref 12, as correctly pointed out by Reviewer 3) were obtained on exponentially growing conditions, a situation in which normally 2 to 5% of mitotic MDAH cells/field could be observed. Thus in this conditions the intrinsic variability of the obtained results hampered the possibility to proper detect subtle differences that might exist. Conversely, here by using synchronization approaches, we were able to more precisely evaluate cell cycle progression and eventually count the differences in cell proliferation. Moreover, we also used single cell clones that are likely more easy to be evaluated. Yes, your question is correct; we expected to find more mitotic cells likely because an altered mitotic division takes longer to be completed.

Moreover, also based on Your suggestion, we included in the new figure S2 the FACS analyses of DNA content of parental and PT-res clones both under exponentially growing condition and after cell synchronization. As expected the bigger differences in cell cycle distribution between parental and PT-Res cells were observed 24 hours after the release from double thymidine block with a higher percentage of G2/M population (13% in parental and 26% or 27% in PT-res clones). Of note, a slightly higher proportion of G2/M population could also be observed in PT-res cells under exponentially growing condition, that however become of some significance only because confirmed in other experimental approaches. These data are now reported and described in the new version of the manuscript at lines  262-265.

10) Section 3.3 - lines 254-267, using isogenic p53 wild type and p53-null cancer cell clones, loss of p53 has previously been shown to promote abnomal cell ploidy, elevated S10P H3 and perturbed progression through M phase after release from nocodazole-induced arrest (Allison & Milner, 2003, Cancer Research 63:6674-9) - this study is relevant and should be cited.

We thank Reviewer 3 for this comment, in the present revised version of our manuscript we have included this reference (lines 304-306).

11) comment on role of S15P p53 given sustained levels observed in Fig. 3d with cisplatin washout in PT-res clones cf. parental cells.

The analyses of TP53 structure (see below an example of the 3D structure of p53 DNA binding domain) showed that S185 is juxtaposed to R273. Therefore, we believed that the substitution of S185G, at least in the context of p53R273H mutant protein, could alter its 3D conformation and capacity to bind the DNA eventually resulting in higher phosphorylation and/or activity. We added this consideration in the discussion section as requested (see lines 437-439).

In the figure is reported the 3D structure of TP53 DNA binding domain in which on the left is marked the position of R273 (green dot) and on the right the position of S185 (blue dot).

12) Following overexpression of the R273H/S185G p53 double mutant in the SKOV3 cells compared to control or R273H cells, is a reduction in growth rate observed? Similarly, have the authors analysed whether there any effects on mitotic fidelity? Is there any change in expression of CCNE2? i.e. the data using the SKOV3 indicates that the secondary mutation of p53 promotes PT-resistance, however, mechanistically is this double mutation causally associated in any way to the mitotic-related observations made in the first half of the manuscript - is there any supportive data from the SKOV3 transfections that can be presented that shows effects similar to those in the MDAH P-res clones? This would add value to the manuscript and the strength of conclusions that can be made eg. lines 422-423

We thank Reviewer 3 for this suggestion that helped us to improve the manuscript. We now show that p53S185G/R273H increased the number of multinucleated cells and decreased the growth of SKOV-3 cells (New Figure 5c and 5e).

We also have investigated the expression some of these mitotic regulators in SKOV3 cells over expressing p53R273H and the novel double mutation p53S185G/R273H. In particular the new Figure 5d shows that DEPDC1 and NCAPH mRNA expression is differentially modulated by p53R273H and p53S185G/R273H in SKOV-3 cells. However, we also evaluated in the same cells CCNE2 expression showing no difference, suggesting that context dependent effects in gene regulation by mutant TP53 exist. These data are now included in the manuscript and commented at lines 422-428. We have also completed the WB analysis in Figure 5a adding the expression of NCAPH and p53 pSer15.

Overall these new data support the possibility that p53S185G/R273H is at least partially responsible of the observed phenotype. This possibility is now commented in the discussion section at lines 459-461. 

13) Comment on pathophysiological relevance - does this secondary mutation in p53 occur in vivo in ovarian cancer or any other cancers?

p53S185G is a rare mutation with only two cases reported in the literature so far, one case of endometrioid ovarian cancer (the same histotype from which MDAH cells derive) and on case of esophageal cancer. In both cases it was present in tumors also carrying other TP53 pathogenic mutation. This point is now clarified (lines 438-444) in the discussion section.

Please see attached files with figures

Round 2

Reviewer 1 Report

Although they did not show whether p53 is required for the establishment of the PT-resistant and MNGCs phenotypes, their results suggested that the expression of mutant TP53 is essential for the survival of MDAH PT-res cells.

Author Response

We thank Reviewer 1 for appreciating and highlighting our work.

Reviewer 2 Report

All issues are properly addressed.

Author Response

We thank Reviewer 2 for appreciating our work.

Reviewer 3 Report

A few minor points as detailed below - but otherwise I would consider acceptable for publication.

The manuscript has been improved in response to reviewers' comments with the inclusion of some new data linking the p53 double mutation to multinucleation and reduced growth in SKOV3 and text/figure alterations providing improved clarity. Most of the reviewers' comments are now adequately addressed for publication in Cells, however a few minor points still remain (easily addressable) before it is acceptable for publication:-

a) Reviewer Point 1 - The trypan blue %s are now included in Fig. S1b as requested. However, this reviewer is less sure of your interpretation in the text  - line 205 - "likely due to increased cell death". The %s are very low - so is this an accurate statement - given the large fold effects on cell growth rate? Please qualify/consider text statement.

b) Reviewer Point 2 - altered morphology is not commented on at all - the significance of this may be unknown/speculative - are the platinum resistant lines more mesenchymal in morphology and potentially more migratory? Include a comment in the text.

c) Reviewer Point 4 - It should be made clear in the methods or figure legend about the gating you describe in the point by point response. I would suggest that the FSC vs SSC and FL2-A vs FL2-W blots be included in the supplementary data - it is apparent there is a modest increase in FSC indicative of increased cell size in clone 42 - could this be shown on an overlay plot/histogram of FSC? Could some of these be gated and shown on a FL2-A vs FL2-W plot to be of higher DNA content than G2?

Author Response

A few minor points as detailed below - but otherwise I would consider acceptable for publication.

The manuscript has been improved in response to reviewers' comments with the inclusion of some new data linking the p53 double mutation to multinucleation and reduced growth in SKOV3 and text/figure alterations providing improved clarity. Most of the reviewers' comments are now adequately addressed for publication in Cells, however a few minor points still remain (easily addressable) before it is acceptable for publication:-

We thank Reviewer 3 for highlighting and appreciating our efforts.

a) Reviewer Point 1 - The trypan blue %s are now included in Fig. S1b as requested. However, this reviewer is less sure of your interpretation in the text  - line 205 - "likely due to increased cell death". The %s are very low - so is this an accurate statement - given the large fold effects on cell growth rate? Please qualify/consider text statement.

We agree with Reviewer 3. The % of dead cells are higher in PT-Res than in parental but likely they cannot explain alone the reduced growth rate observed in the latter cells. We better clarify point in this revised manuscript at lanes 205-206.

b) Reviewer Point 2 - altered morphology is not commented on at all - the significance of this may be unknown/speculative - are the platinum resistant lines more mesenchymal in morphology and potentially more migratory? Include a comment in the text.

We agree Reviewer 3 and commented on this point in the discussion section at lanes 491-495.

c) Reviewer Point 4 - It should be made clear in the methods or figure legend about the gating you describe in the point by point response. I would suggest that the FSC vs SSC and FL2-A vs FL2-W blots be included in the supplementary data - it is apparent there is a modest increase in FSC indicative of increased cell size in clone 42 - could this be shown on an overlay plot/histogram of FSC? Could some of these be gated and shown on a FL2-A vs FL2-W plot to be of higher DNA content than G2?

As requested, we have now included these data in the new figure S2d and commented at lines 264-265.